# *ALMS1-IT1*: A Key Player in the Novel Disulfidptosis-Related LncRNA Prognostic Signature for Head and Neck Squamous Cell Carcinoma

**DOI:** 10.3390/biom14030266

**Published:** 2024-02-23

**Authors:** Xin-Yi Sun, Mian Xiao, Min Fu, Qian Gao, Rui-Feng Li, Jing Wang, Sheng-Lin Li, Xi-Yuan Ge

**Affiliations:** 1Central Laboratory, Peking University School and Hospital of Stomatology, Beijing 100081, China; 2National Center for Stomatology & National Clinical Research Center for Oral Diseases & National Engineering Research Center of Oral Biomaterials and Digital Medical Devices, Beijing 100081, China; 3Beijing Key Laboratory of Digital Stomatology, Beijing 100081, China; 4Department of Oral and Maxillofacial Surgery, Peking University School and Hospital of Stomatology, Beijing 100081, China

**Keywords:** disulfidptosis, head and neck squamous cell carcinoma (HNSCC), lncRNA, *ALMS1-IT1*, pentose phosphate pathway (PPP), NADPH

## Abstract

Disulfidptosis is a newly discovered form of programmed cell death that is induced by disulfide stress. It is closely associated with various cancers, including head and neck squamous cell carcinoma (HNSCC). However, the factors involved in the modulation of disulfidptosis-related genes (DRGs) still remain unknown. In this study, we established and validated a novel risk score model composed of 11 disulfidptosis-related lncRNAs (DRLs) based on 24 DRGs in HNSCC. The results revealed strong correlations between the 11-DRL prognostic signature and clinicopathological features, immune cell infiltration, immune-related functions, and disulfidptosis-associated pathways, including NADPH and disulfide oxidoreductase activities. Furthermore, we studied and verified the involvement of *ALMS1-IT1*, one of the 11 model DRLs, in the disulfidptosis of HNSCC cell lines. A series of assays demonstrated that *ALMS1-IT1* modulated cell death under starvation conditions in a pentose phosphate pathway (PPP)-dependent manner. Knockdown of *ALMS1-IT1* inhibited the PPP, contributing to a decline in NADPH levels, which resulted in the formation of multiple intermolecular disulfide bonds between actin cytoskeleton proteins and the collapse of F-actin in the cytoplasm. Therefore, *ALMS1-IT1*, which is highly expressed in *SLC7A11*^high^ cells, can be considered a promising therapeutic target for disulfidptosis-focused treatment strategies for cancer and other diseases.

## 1. Introduction

Head and neck squamous cell carcinoma (HNSCC), the sixth most prevalent malignant tumor worldwide, is an aggressive and recurrent neoplasm. It is characterized by a high morbidity and poor prognosis [1]. Extensive research has been dedicated to elucidating the etiology and pathogenesis of HNSCC, revealing the multifaceted contributions of metabolism, immunity, angiogenesis, and the tumor microenvironment [2].

Disulfidptosis is a recently described form of programmed cell death distinct from other modes of cell death [3,4,5]. Triggered by glucose starvation in *SLC7A11*^high^ cells, disulfidptosis is characterized by the actin cytoskeleton due to an aberrant accumulation of intracellular disulfides. The stimulation of disulfidptosis may require three key factors: (1) high expression of *SLC7A11*, a cystine transporter that uptakes extracellular cystine at high rates and reduces it to cysteine through an NADPH-consuming reaction; (2) glucose deprivation, which depletes the cytosolic NADPH pool that is mainly generated from the pentose phosphate pathway (PPP); and (3) disulfide stress, which induces cell death by altering the conformation of cytoskeletal proteins [4,5,6]. Distinct from apoptosis and necroptosis, which require specific protein machineries, no factor directly involved in the execution of disulfidptosis has been identified thus far. While current research has shed light on the potential of disulfidptosis in tumor treatment, its underlying mechanism and potential therapeutic applications in HNSCC require further study.

The intra- and intertumor heterogeneity of HNSCC contribute to poor treatment outcomes [7]. Therefore, targeting a single gene may not yield the desired effect. A model that integrates multiple genes as biomarkers of HNSCC is thus urgently needed. Long noncoding RNAs (lncRNAs) are a class of RNA molecules longer than 200 nucleotides that do not encode proteins [8]. Compared with messenger RNAs (mRNAs) or microRNAs (miRNAs), accumulating evidence suggests that lncRNAs play crucial roles in various cellular processes, including gene expression regulation, chromatin modification, and cellular differentiation [9]. Thus, a comprehensive evaluation of multiple lncRNAs may be useful in the prognostic assessment of patients with HNSCC.

In this study, we constructed a prognostic model based on 44 normal and 522 HNSCC samples using disulfidptosis-related lncRNAs to predict the prognosis, recurrence, and survival of patients. Eleven lncRNAs were found to participate in disulfidptosis and affect the clinical outcomes of patients with HNSCC. Among these 11 model lncRNAs, we found that *ALMS1-IT1* plays an important role in disulfidptosis, being generally associated with advanced disease and poor clinical outcomes. The current findings may help optimize precision treatment and provide new therapeutic targets.

## 2. Materials and Methods

### 2.1. Data Collection and Processing

RNA-seq data and clinical information of TCGA Pan-Cancer Atlas and TCGA-HNSC were acquired from The Cancer Genome Atlas Program (TCGA) database (https://portal.gdc.cancer.gov/ (accessed on 29 March 2023)). Ensembl IDs were converted to Gene Symbols, and log2 processing of the RNA-seq data was performed. Protein-coding genes and lncRNAs were screened by Ensembl human genome browser GRCh38.

### 2.2. Identification of Disulfidptosis-Related lncRNAs

The list of disulfidptosis-related genes (DRGs) was acquired from previous studies [3,4]. Pearson correlation coefficients were calculated according to DRGs and lncRNA expression profiles to identify disulfidptosis-related lncRNAs (|r| > 0.3, *p* < 0.001).

### 2.3. Establishment and Verification of a Disulfidptosis-Related lncRNA Prognostic Model

The construction and validation of a disulfidptosis-related lncRNA (DRL) prognostic signature were conducted as previously described [10,11,12,13]. The DRLs were filtered via a univariate Cox regression analysis. A total of 519 patients with HNSCC were randomly divided into training or test cohorts at a ratio of 1:1. Subsequently, a univariate/multivariate Cox regression analysis and least absolute shrinkage and selection operator (LASSO) regression analysis were applied to establish a prognostic model. The formula for the risk score for each patient was:(1)$$Risk\;score=\sum_{i=0}^{n}coef_{i}\times x_{i}$$
where $x_{i}$ and $coef_{i}$ represent the expression of each lncRNA and its corresponding coefficient, respectively.

According to the median value of the risk scores, the patients in the training, test, and overall cohorts were classified into low-risk and high-risk groups. The Kaplan–Meier survival curve with log-rank tests was utilized to compare the overall survival (OS) between the low- and high-risk groups. A receiver operating characteristic (ROC) curve was utilized to evaluate the predictive accuracy of the prognostic signature.

### 2.4. Functional Enrichment Analysis and Gene Set Enrichment Analysis

The differentially expressed genes (DEGs) between the high-risk and low-risk groups were identified (|log_2_ (Fold Change)| > 1, FDR < 0.05) and functionally annotated according to Gene Ontology (GO) and Kyoto Encyclopedia of Genes and Genomes (KEGG) enrichment analyses (adjusted *p*-value < 0.05). A gene set enrichment analysis (GSEA) was performed to explore the molecular and biological differences between the high-risk and low-risk groups (*p* < 0.05, FDR < 0.25).

### 2.5. Assessment of Immune Cell Infiltration and Immune Microenvironment

The difference of immune cell infiltration between the high-risk and low-risk groups was evaluated using TIMER, CIBERSORT, CIBERSORT-ABS, quanTIseq, MCP-counter, xCell, and EPIC algorithms.

### 2.6. Cell Lines and Culture

The HN-6 cell line was obtained from Ninth People’s Hospital, Shanghai Jiao Tong University School of Medicine (Shanghai, China). The HOK cell line was purchased from Shanghai Enzyme-linked Biotechnology Co., Ltd. (Shanghai, China). Cells were cultured in Dulbecco’s modified Eagle medium (DMEM) with 10% (*v*/*v*) fetal bovine serum (FBS) and 1% (*v*/*v*) penicillin-streptomycin in a 5% CO_2_ air atmosphere at 37 °C. For glucose deprivation experiments, cells were cultured in glucose-free DMEM with 10% FBS, as previously described [3,11]. The glucose-free DMEM was obtained from Procell Life Science & Technology (PM150270) (Wuhan, China).

### 2.7. Small Interfering RNA (siRNA) Transfection

All siRNAs were designed and synthesized by RiboBio (Guangzhou, China). Cells were seeded in 6-well plates 18–24 h before transfection. When the confluence was up to 30–50%, siRNAs were distributed to cells using a TransIT-X2^®^ Dynamic Delivery System (MIR 6000; Mirus Bio, Madison, WI, USA) as per the manufacturer’s instructions. Approximately 48 h after transfection, cells were harvested for further experiments. The siRNAs used in this study are listed in Appendix A.

### 2.8. RNA Extraction and Reverse Transcription–Quantitative Real-Time Polymerase Chain Reaction (RT-qPCR)

Total cellular RNA was extracted using the SteadyPure Quick RNA Extraction Kit (AG21023; Accurate Biotechnology, Changsha, China). The obtained RNA was quantified on a NanoDrop™ 8000 Spectrophotometer (Thermo Fisher Scientific, Waltham, MA, USA) and then used for complementary DNA (cDNA) synthesis with an *Evo M-MLV* RT Mix Tracking Kit with gDNA Clean for qPCR (Yellow) (AG11734; Accurate Biotechnology). Quantitative real-time PCR was conducted using an SYBR Green Premix *Pro Taq* HS qPCR Tracking Kit (Rox Plus) (AG11735; Accurate Biotechnology) and a QuantStudio™ 3/6 Flex System (Applied Biosystems, Foster City, MA, USA). The gene expression levels were calculated using the 2−∆∆Ct method. The primer sequences used for RT-qPCR are listed in Appendix A.

### 2.9. Cell Death Assays

Cell death assays were performed as previously described [3]. To measure the proportion of live and dead cells within a sample population, cells were grown in 6-well plates before treatment. After being transfected with siRNAs and cultured in glucose-free media, the cells were digested with Accutase^®^ Cell Detachment Solution (423201; BioLegend, San Diego, CA, USA) and collected in 5 mL test tubes (352058; Falcon, Marlboro, NY, USA). After two washes with cold PBS, the cells were resuspended in 10 μL/mL PI in 1× Binding Buffer (556547; BD Biosciences, Franklin Lakes, NJ, USA), incubated for 15 min at RT (25 °C) in the dark, and then analyzed via flow cytometry within 1 h. Dead (PI-positive) cells were analyzed using a BD FACSCalibur 2 (BD Biosciences) and FlowJo v10 software.

### 2.10. Western Blot

Western blotting was performed according to standard protocols, with special modifications for protein preparation [14]. NuPAGE^®^ LDS sample buffer (4×; NP0007; Life Technologies, Waltham, MA, USA), which was free from reducing agents, was used to prepare samples. Protein samples were divided into two aliquots, one of which was treated with NuPAGE^®^ LDS sample buffer alone as the non-reduced sample, while the other was treated with BeyoGel™ sample reducing agent (10×; P0733; Beyotime, Shanghai, China) in addition to NuPAGE^®^ LDS sample buffer as the reduced sample. Both reduced and non-reduced samples were heated at 70 °C for 10 min for denaturing gel electrophoresis. The antibodies used for Western blotting are listed in Appendix A. Protein bands were imaged using an e-BLOT electronic compression imaging system (Touch Imager; e-BLOT, Shanghai, China).

### 2.11. NADP^+^ and NADPH Measurements

To calculate the ratio of NADP^+^/NADPH, the intracellular levels of NADPH and total NADP (NADPH + NADP^+^) were measured using an NADP^+^/NADPH Assay Kit with WST-8 (S0179; Beyotime). NADP^+^ and NADPH measurement assays were performed according to the manufacturer’s instructions, with optimization. First, the cells were grown in 12-well plates, transfected with siRNAs, and cultured in glucose-free medium. Then, the cells (about 5 × 10^5^ cells/well) were lysed with 400 μL of NADP^+^/NADPH extraction buffer, collected into 1.5 mL microtubes, and centrifuged at 12,000× *g* and 4 °C. The supernatant in each tube was split into two 200 μL aliquots. For the detection of NADPH, one aliquot was incubated at 60 °C for 30 min in a metal bath to deplete NADP^+^. The other aliquot was used to detect the total NADP (NADP_total_). Next, 50 μL samples from both aliquots were added into a 96-well plate along with 100 μL of G6PDH working buffer and incubated at 37 °C for 10 min in the dark. After the addition of 10 μL coloring solution to each well, the plate was incubated at 37 °C for 10–20 min. The absorbance of each well was measured at 450 nm using a microplate reader. Based on the standard curve established in the assay, the concentrations of NADP^+^ and NADPH were subsequently calculated via the following two formulas: [NADP^+^] = [NADP_total_] − [NADPH]; [NADP^+^]/[NADPH] = ([NADP_total_] − [NADPH])/[NADPH].

### 2.12. Immunofluorescence Staining of Actin Filaments

Immunofluorescence was performed as previously described [15]. The cells were grown in 20 mm glass bottom cell culture dishes (801001; NEST Biotechnology, Wuxi, China). After culturing in special media with or without the appropriate drugs, the cells were gently washed twice with ice-cold PBS for 5 min per wash, fixed by adding a volume of 4% paraformaldehyde (P0099; Beyotime) for 20 min, and permeabilized with PBS containing 0.1% Triton X-100 for 10 min. Next, the cells were stained with Actin-Tracker Red-594 (C2205S; Beyotime) in Immunofluorescence Staining Secondary Antibody Dilution Buffer (P0108; Beyotime) at a 1:100 dilution, then incubated at room temperature in the dark for 1 h, followed by two washes with PBS containing 0.1% Triton X-100. Subsequently, the dishes and coverslips were mounted with Antifade Mounting Medium with DAPI (P0131; Beyotime) and allowed to dry in the dark before viewing. All fluorescence images were captured using a confocal microscope (LSM 710; ZEISS, Oberkochen, Germany).

### 2.13. Statistical Analysis

All assays were independently repeated at least twice, and similar results were obtained. All data are presented as means ± standard deviation (SD) or means with 95% CIs. All bioinformatic analyses were performed using R software (v4.3.2). All statistical analyses were performed using GraphPad Prism software (v10). *p* < 0.05 indicates a statistically significant difference. * represents *p* < 0.05, ** represents *p* < 0.01, and *** represents *p* < 0.001. ns represents not significant (*p* ≥ 0.05).

## 3. Results

### 3.1. Publicly Available Data Collection and Identification of Disulfidptosis-Related lncRNAs in HNSCC

The flowchart presents the basic process framework of this study (Figure 1). As disulfidptosis is a new type of cell death that occurs in *SLC7A11*^high^ cells, we screened cancer types with a high *SLC7A11* expression [3,5]. Based on data from the TCGA Pan-Cancer Atlas, among a total of 33 cancer types, there were 15 cancer types with *SLC7A11* upregulation in tumor samples compared to normal samples in both paired and unpaired data, including BRCA, CHOL, COAD, ESCA, HNSC, KICH, KIRC, KIRP, LIHC, LUAD, LUSC, PAAD, PRAD, READ, and STAD (Figure 2a,b). In the present study, we focused on HNSCC.

Gene expression profiles and clinical data of 44 normal and 522 HNSCC samples were obtained from the HNSCC dataset of TCGA (TCGA-HNSC). Based on previous studies, 24 genes were identified as disulfidptosis-related genes (DRGs) [3,4]. To further explore candidate disulfidptosis-related lncRNAs (DRLs), a Pearson’s correlation analysis was used to evaluate the correlation between the expression levels of the 24 DRGs and those of lncRNAs (|r| > 0.3, *p* < 0.001). Subsequently, we found that 1451 lncRNAs were related to disulfidptosis in HNSCC (Figure 2c).

### 3.2. Construction of an 11-DRL Signature as a Novel Prognostic Model for HNSCC

To evaluate the prognostic value of these DRLs, TCGA-HNSC samples were randomly divided into training and test groups. There were no significant differences in clinical characteristics between the two groups (Appendix A).

A univariate Cox regression analysis was used to identify 79 DRLs as prognostic factors according to survival time in the training group (Appendix A). A least absolute shrinkage and selection operator (LASSO) regression analysis was used to simplify the predictive model for greater accuracy (Figure 2d,e). A disulfidptosis-related prognostic risk evaluation model based on 11 DRLs, including *KLHL7-DT*, *AC007066.3*, *LINC01123*, *AC087752.4*, *AL162458.1*, *LINC01281*, *AC245041.2*, *SNHG7*, *AC108010.1*, *AL109936.2*, and *ALMS1-IT1*, was ultimately constructed using a multivariate Cox regression analysis. According to this prognostic signature, each HNSCC patient in the TCGA was assigned a risk score, calculated as per the following formula: risk score = 0.4006 × *KLHL7-DT* + 0.5354 × *AC007066.3* + 0.7787 × *LINC01123* + (−0.4349) × *AC087752.4* + (−1.0036) × *AL162458.1* + (−0.9560) × *LINC01281* + 0.3306 × *AC245041.2* + 0.3284 × *SNHG7* + (−0.6826) × *AC108010.1* + (−0.3378) × *AL109936.2* + 0.6325 × *ALMS1-IT1*. A further correlation analysis revealed that the 11 model DRLs interacted closely with each other (Figure 2f,g), and there was a strong association between the DRGs and DRLs (Figure 2h).

### 3.3. Validation and Evaluation of the 11-DRL Prognostic Signature in HNSCC

To validate the prognostic potential of this 11-DRL signature, samples from the training group were classified into low- and high-risk clusters based on the median risk score value. Kaplan–Meier estimates confirmed that the high-risk cluster was associated with poorer overall survival (OS) and progression-free survival (PFS) than the low-risk cluster (Figure 3a–f). As depicted in the scatter plots, in the overall group, survival time and living status were ranked by the distribution of risk scores, and the mortality rates of patients with HNSCC increased with the risk score (Figure 3g,h). Additionally, the heatmap of the 11 model DRLs revealed that in the high-risk cohort, the expression levels of *KLHL7-DT*, *AC007066.3*, *LINC01123*, *AC245041.2*, *SNHG7*, and *ALMS1-IT1* were markedly upregulated, whereas those of *AC087752.4*, *AL162458.1*, *LINC01281*, *AC108010.1*, and *AL109936.2* were apparently lower (Figure 3i and Appendix A). To further assess the predictive power of this model, double validations were performed on the K m survival curves, distribution figures, and heat maps for the training and test groups (Appendix A).

We evaluated the relationship between the 11-DRL prognostic signature and the clinicopathological features of HNSCC. Six lncRNAs (*KLHL7-DT*, *AC007066.3*, *LINC01123*, *AC245041.2*, *SNHG7*, and *ALMS1-IT1*) in the signature were considered risk lncRNAs, as their expression levels were upregulated in the high-risk cohort. The other five lncRNAs (*AC087752.4*, *AL162458.1*, *LINC01281*, *AC108010.1*, and *AL109936.2*) were deemed protective indicators with a decreased expression in the same cohort. Remarkably, the high-risk cohort exhibited a shorter survival time and more advanced clinical and T stages than the low-risk cohort (Figure 3i–k). Additionally, a UMAP was used to confirm the clustering efficacy of the signature in distinguishing the survival risk of patients with HNSCC (Figure 3l). In brief, these results indicate that the 11-DRL signature could categorize risk levels and evaluate the prognosis of HNSCC patients through risk-score-based predictions.

To determine the accuracy of the 11-DRL risk score as an independent prognostic factor in HNSCC, we subjected it and other indicators to univariate and multivariate Cox regression analyses. The results revealed that the 11-DRL risk score could serve as an independent prognostic factor to predict the survival rates of patients with HNSCC, in addition to age, N stage, and M stage (Figure 4a).

Subsequently, a nomogram for the prediction of the 1-year, 3-year and 5-year overall survival was built by calculating the sum of prognostic factors, including age, sex, clinical stage, TNM stage, and risk score (Figure 4b). The calibration curve of the nomogram showed good consistency between the predicted results and real-world clinical records (Figure 4c).

Moreover, a time-dependent receiver operating characteristic (ROC) analysis suggested the prognostic performance of the established model (Figure 4d). The prognostic accuracies of the nomogram and signature were appraised using the area under the curve (AUC) and through a comparison with other clinicopathological information (Figure 4e). The results revealed that nomogram and risk score performances were superior to those of other indicators. Interestingly, the C-index curves indicated that the performance of the nomogram preceded the risk score for the prediction of 1-year and 2-year overall survival. However, over time, there was no significant difference between the nomogram and risk score (Figure 4f). Cumulatively, these data show that the 11-DRL model could serve as an independent prognostic tool for HNSCC.

### 3.4. Dysregulated Immunity-, NADPH-, and Disulfide-Related Pathways between the DRL-Based Risk Clusters

To investigate the underlying dysregulated biological processes and signaling pathways in the two risk groups classified based on the 11-DRL prognostic model, we performed Gene Ontology (GO) and Kyoto Encyclopedia of Genes and Genomes (KEGG) enrichment analyses and found that the differentially expressed genes (DEGs) between the low- and high-risk groups were mainly involved in immunity-related pathways (Figure 5a,b). A gene set enrichment analysis (GSEA) revealed that the high-risk cluster exhibited ribosome hyperfunction and an insufficient immune function (Figure 5c). To reduce the redundancy of pathway enrichment analysis (PEA) results, we leveraged the similarities between pathway gene sets and represented them as networks of interconnected clusters. Each cluster was assigned a meaningful name reflecting its biological significance, which enabled an objective overview of the data [16] (Figure 5d). Interestingly, the GSEA and GO enrichment analysis revealed that the DEGs were notably associated with NADPH-related pathways and disulfide oxidoreductase activities, which play pivotal roles in disulfidptosis (Figure 5e). The core genes, some of which could be considered promising therapeutic targets for HNSCC, were annotated at the leading edges of the four enriched signaling pathways (Figure 5f–i). Thus, dysregulated immunity-related pathways may underpin the poor prognosis of the high-risk HNSCC group based on the 11-DRL model.

### 3.5. Immune Landscape in HNSCC and Its Association with the 11-DRL Risk Score

To compare immune cell infiltration between the high and low DRL score groups, several analytical tools, including TIMER, CIBERSORT, CIBERSORT-ABS, quanTIseq, MCP-counter, xCell, and EPIC, were employed to estimate the abundance of immune, stromal, and uncharacterized cells in the tumor microenvironment (TME) (Figure 6a). CIBERSORT, a widely used deconvolution method initially optimized for immune cell types, consistently outperformed other algorithms [17,18]. Thus, we used it to assess the association between the DRL risk score and immune cell infiltration based on immunity-related genes. Remarkably, M2-like tumor-associated macrophages, which promote immune suppression and facilitate tumor evasion in neoplastic diseases [19], exhibited greater infiltration in the high DRL score group. In addition, there was greater infiltration of plasma cells and activated mast cells, which exerted both positive and negative immune regulation [20,21]. Intriguingly, T follicular helper (Tfh) cells, a greater frequency of which is often associated with a better prognosis in solid tumors of non-lymphocytic origin [22], were more deficient in the TME of the high-risk group, whereas immunosuppressive cells, such as regulatory T cells (Tregs) [23], were more abundant (Figure 6b). Given that the high DRL score implied a poor survival, it seems that the infiltration of immune cells could not directly explain this. Thus, to determine the immune status in low- and high-risk groups more accurately, we shifted our attention to immune-related functions. The scores of 29 immune-related functions generally decreased in the high-risk group, especially those of CD8^+^ T cells, Tfh cells, mast cells, NK cells, type II IFN response, and checkpoints (Figure 6c). The expression levels of classic co-stimulatory factors of T and NK cells, such as *TNFRSF4 (OX40)*, *TNFRSF9 (4-1BB)*, *ICOS*, *CD27*, *CD28*, *CD40LG (CD40L)*, and *CD226* [24], were remarkably downregulated, while *CD276 (B7-H3)*, which suppresses T and NK cell activation [25,26], was upregulated in the high-risk group (Figure 6d,e). As described above, these results suggest that a high DRL score is associated with a more compromised immune status, which could partially account for the poorer prognosis of the high-risk group.

### 3.6. High Expression of ALMS1-IT1 Is Associated with Poor Prognosis in HNSCC, Modulating Cell Death under Starvation in a PPP-Dependent Manner

To filter out and further validate the potential targets related to disulfidptosis in HNSCC, we performed a K–M survival analysis between the low- and high-risk groups based on the 11-DRL model. The results indicated that seven lncRNAs could be regarded as independent prognostic factors (*p* < 0.05), of which two lncRNAs (*LINC01123* and *ALMS1-IT1*) acted as risk factors, and the other five lncRNAs (*AC087752.4*, *AL162458.1*, *LINC01281*, *AC108010.1*, and *AL109936.2*) served as protective factors (Appendix A). Additionally, the expression levels of the 11 model DRLs were compared between normal and tumor samples from TCGA-HNSC. Both independent risk prognostic lncRNAs (*LINC01123* and *ALMS1-IT1*) were significantly upregulated in tumor samples, while only one lncRNA (*AL109936.2*) of the five independent protective prognostic indicators was downregulated (Appendix A). These observations were consistent with the tendencies observed between the low- and high-risk cohorts.

Although *ALMS1-IT1* has been reported as a novel tumor promoter that may facilitate the malignant progression of LUAD [27], there is a lack of evidence supporting its involvement in cancer disulfidptosis. To confirm the occurrence of disulfidptosis in HNSCC and the participation of *ALMS1-IT1*, we first verified the high expression levels of *SLC7A11* and *ALMS1-IT1* in HNSCC cell lines. *SLC7A11* and *ALMS1-IT1* were simultaneously overexpressed in HN-6 cells (Figure 7a,b). To gain insights into the biological functions of *ALMS1-IT1* during disulfidptosis in HN-6 cells, we detected the expression of genes related to the PPP, which contributes most to intracellular NADPH generation, supporting the reduction of disulfide bonds [4,28]. Upon *ALMS1-IT1* knockdown, most PPP-related genes were downregulated (Figure 7c,d).

We then determined the proportion of HN-6 cells that died after *ALMS1-IT1* silencing under glucose starvation. Previous research demonstrated that disulfide stress was the cause of cell death in disulfidptosis; that is, the formation of disulfide bonds and changes in the NADP^+^/NADPH ratio preceded obvious cell death in *SLC7A11*^high^ cells [3,4]. We observed obvious cell death (>20%) approximately 8 h after glucose starvation, indicating that the downregulation of *ALMS1-IT1* might accelerate cell death under glucose deprivation (Figure 7e–g and Appendix A). Moreover, knockdown of *ALMS1-IT1* resulted in a significant increase in the NADP^+^/NADPH ratio, particularly during glucose deprivation (Figure 7h). These data indicate that *ALMS1-IT1* plays a vital role in disulfidptosis by regulating the PPP pathway, which might lead to a poorer HNSCC prognosis.

### 3.7. ALMS1-IT1 Regulates Disulfide Bond Formation between Actin Cytoskeleton Proteins during Disulfidptosis

Next, we sought to validate disulfide bond formation between the identified actin cytoskeleton proteins during disulfidptosis before obvious cell death. Notably, under non-reducing conditions with the preservation of disulfide bonds, multiple actin cytoskeleton proteins exhibited slower migration with smears and extremely high-molecular-weight bands under glucose starvation in wild-type (WT) HN-6 cells. With prolonged glucose starvation, multiple actin cytoskeleton protein bands deepened and were gradually enlarged under non-reducing conditions. In contrast, no such migration was observed under reducing conditions, indicating the formation of multiple intermolecular disulfide bonds between actin cytoskeleton proteins under glucose starvation (Figure 8a).

Furthermore, *ALMS1-IT1* knockdown resulted in a similarly slower migration of actin cytoskeleton proteins to that observed under glucose deprivation and non-reducing conditions. Multiple actin cytoskeleton proteins with smears and extremely high-molecular-weight bands became more apparent following glucose starvation, *ALMS1-IT1* knockdown, or both in HN-6 cells (Figure 8b). Collectively, these data suggest that *ALMS1-IT1* silencing significantly promotes disulfide bond formation between actin cytoskeleton proteins, which could ultimately lead to disulfidptosis.

### 3.8. ALMS1-IT1 Modulates F-Actin Collapse during Disulfidptosis

Actin filaments (F-actin) were stretched out in the cytoplasm of HN-6 cells cultured in a glucose-containing medium, as observed via phalloidin staining. Glucose deprivation in WT HN-6 cells induced striking changes in cell morphology, with F-actin collapse and cell contraction, which were aggravated by prolonged glucose starvation (Figure 8c). In addition, *ALMS1-IT1* knockdown exacerbated F-actin contractions under glucose starvation (Figure 8d). In the presence of glucose, the addition of glucose transporter (GLUT) inhibitors, such as BAY-876 and Glutor, caused similar cell shrinkage under glucose-free conditions (Figure 8e). Silencing *ALMS1-IT1* further suppressed the contraction of F-actin. *ALMS1-IT1* knockdown suppressed NADPH production, showing a synergistic effect with glucose deprivation (Figure 8e). In contrast, 2-DG, a glucose analog, exerted the opposite effects on cell morphology. F-actin collapse was abolished by 2-DG treatment, which rescued NADPH generation [29,30,31] (Figure 8f). In addition, TCEP had a similar effect in rescuing cell shrinkage as a widely used reducing agent, which generates a stronger S-S reductive reaction than other chemical reductants [32] (Figure 8f). Strikingly, the rescue effect of 2-DG was superior to that of TCEP after *ALMS1-IT1* silencing, implying that the knockdown of *ALMS1-IT1* exerted its effects via NADPH rather than by directly targeting disulfide bonds. All these morphological changes preceded obvious cell death in these cells, suggesting that cell death resulted from aberrant F-actin collapse.

## 4. Discussion

The prognosis of HNSCC depends on various factors, including sex, age, clinical stage, and TNM stage. As HNSCC exhibits considerable heterogeneity in terms of pathogenesis, constructing a prognostic model that accounts for more pathogenic factors is essential. Disulfidptosis, a new form of regulated cell death induced by disulfide stress, promotes disulfide bond formation between actin cytoskeleton proteins and F-actin collapse [3,4]. The significance of this newly discovered form of cell death for HNSCC prognosis remains unexplored. In the present study, we established a novel 11-DRL signature as a prognostic model for HNSCC based on 24 DRGs. When divided into low- and high-risk groups based on the 11-DLR score, HNSCC cases in the high-risk cohort were characterized by a lower survival and advanced clinicopathological features, which highlighted the prognostic value of this disulfidptosis-related signature. Based on the 11-DRL risk score, high-risk patients exhibited a compromised immune function. These results suggested that a high DRL score is associated with a compromised immune status, which could explain the poor prognosis in the high-risk group. However, the current model remains to be validated in the clinic.

At present, the range of DRGs is rather limited. Several recent studies identified a new cluster of DRGs, including *DCBLD2*, *JAM3*, *CSPG4*, *SCEL*, *GOLGA8A*, *CNTN1*, *APLP1*, *PTPRR*, *POU5F1*, *CTSE*, *MSH3*, *CRB3*, *AUP1*, *RNF10*, and *ELF1*, among others [33,34]. As more DRGs are revealed, additional candidate DRLs are also expected to emerge. Taken together, disulfidptosis represents a promising target for tumor therapy that remains to be exploited.

Although a number of disulfidptosis-related lncRNA prognostic signatures can predict the survival of cancer patients [35,36,37], the involvement of individual lncRNAs in disulfidptosis remains to be experimentally validated. This study is the first to identify an individual lncRNA that can be regarded as a regulatory factor in disulfidptosis. That is, we found that *ALMS1-IT1* knockdown inhibits the PPP, accounting for the observed decline in NADPH levels. Under glucose deprivation, an increase in the NADP^+^/NADPH ratio ensues, accelerating disulfide bond formation between actin cytoskeleton proteins, which leads to F-actin contraction in the cytoplasm. These results demonstrate the pivotal role of *ALMS1-IT1* in disulfidptosis, which may exert a crucial influence on the prognosis and therapeutic response of patients with HNSCC. However, the mechanism through which *ALMS1-IT1* suppresses the PPP remains unclear, requiring further study. Our future studies will concentrate on lncRNAs that are directly correlated with *SLC7A11*, a key factor in both disulfidptosis and ferroptosis. Thus, probing the relationship and interaction between these two types of cell death in *SLC7A11*^high^ cells is of great interest.

In our study, we found that *ALMS1-IT1* silencing significantly promoted F-actin collapse, leading to disulfidptosis. As previously reported, we confirmed that GLUT inhibitors can also trigger disulfidptosis, which is of relevance in the treatment of tumors with high *SLC7A11* expression [3]. In our study, inhibitors BAY-876 and Glutor, which selectively target GLUT-1, -2, and -3 [29,30,31], potently promoted F-actin collapse in *SLC7A11*^high^ cells. The effects of pan-GLUT inhibitors, such as DRB18 [38], need to be studied in greater depth. A synergy between GLUT inhibitors and *ALMS1-IT1* silencing may inform disulfidptosis-related therapeutic strategies for cancer and other diseases.

## 5. Conclusions

In our study, through multiple bioinformatic approaches, we constructed and evaluated a disulfidptosis-related prognostic signature for HNSCC. Moreover, we identified a novel disulfidptosis-related lncRNA, *ALMS1-IT1*, which suppresses the PPP to promote a decline in NADPH levels. In synergy with glucose starvation, the increased NADP^+^/NADPH ratio promotes the formation of multiple intermolecular disulfide bonds between actin cytoskeleton proteins, leading to the collapse of F-actin, which could in turn trigger disulfidptosis in *SLC7A11*^high^ cells. In conclusion, the novel 11-DRL signature model can help predict the prognosis of patients with HNSCC, with *ALMS1-IT1* demonstrating potential as a therapeutic target in HNSCC.

## Figures and Tables

**Figure 1 biomolecules-14-00266-f001:**
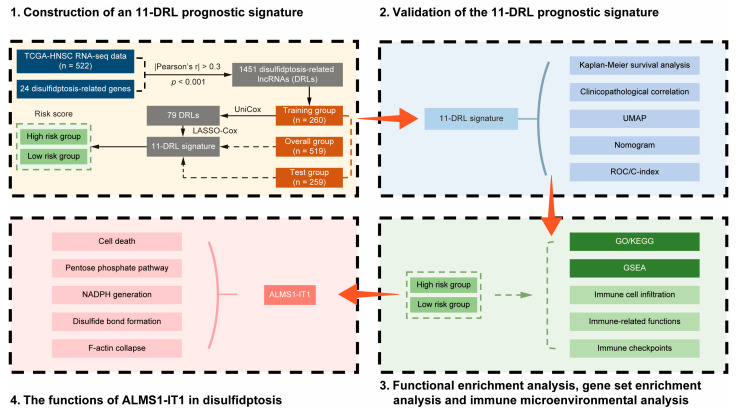
The flowchart of this study.

**Figure 2 biomolecules-14-00266-f002:**
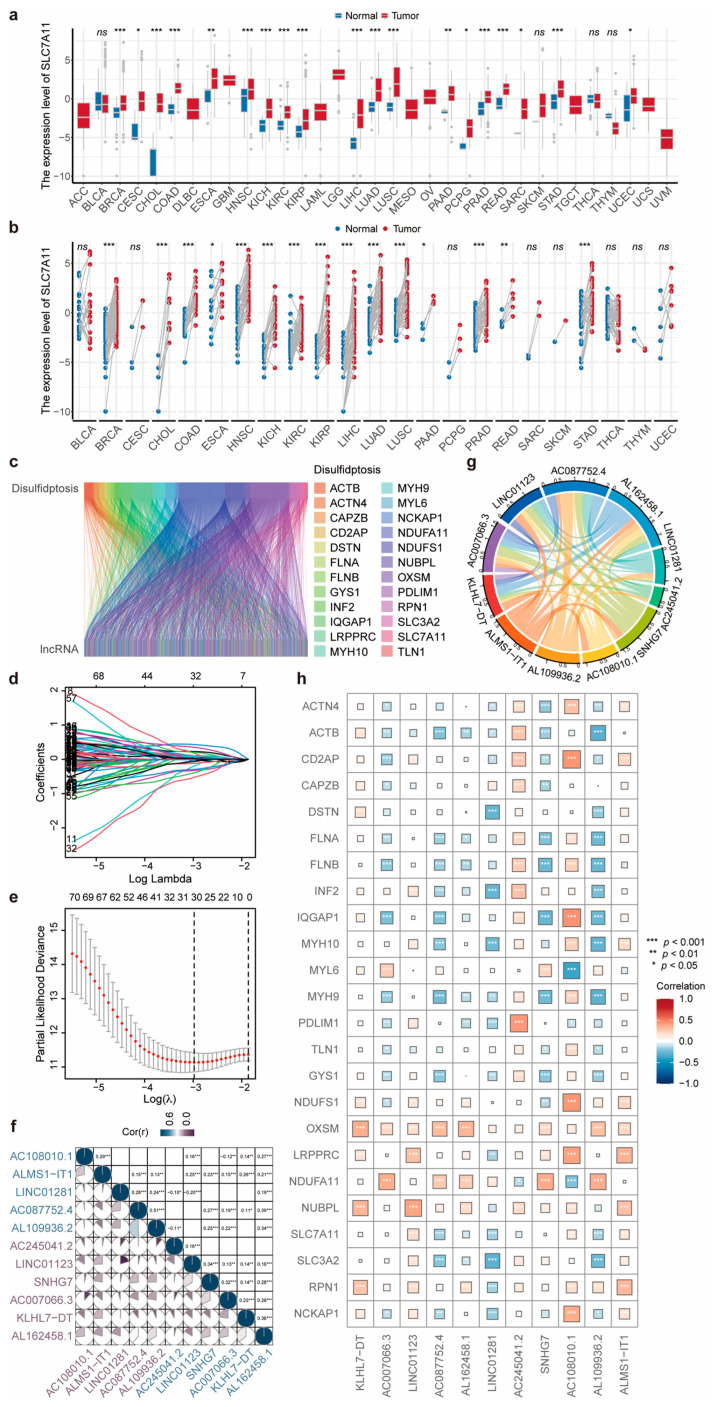
Identification of an 11-DRL prognostic signature. (**a**,**b**) The expression levels of *SLC7A11* in unpaired (**a**) and paired (**b**) tissue samples from the Cancer Genome Atlas (TCGA) database. (**c**) Sankey diagram of lncRNAs associated with disulfidptosis-related genes (|r| > 0.3, *p* < 0.001). (**d**,**e**) The coefficient distribution of LASSO-Cox regression analysis and adjustment parameters were calculated based on partial likelihood deviation and ten-fold cross-validation. (**f**,**g**) Correlation graphs between each of the 11 model DRLs. (**h**) Correlation heatmap between the 11 model DRLs involved in the prognostic model and disulfidptosis-related genes. ns, not significant; *, *p* < 0.05; **, *p* < 0.01; ***, *p* < 0.001.

**Figure 3 biomolecules-14-00266-f003:**
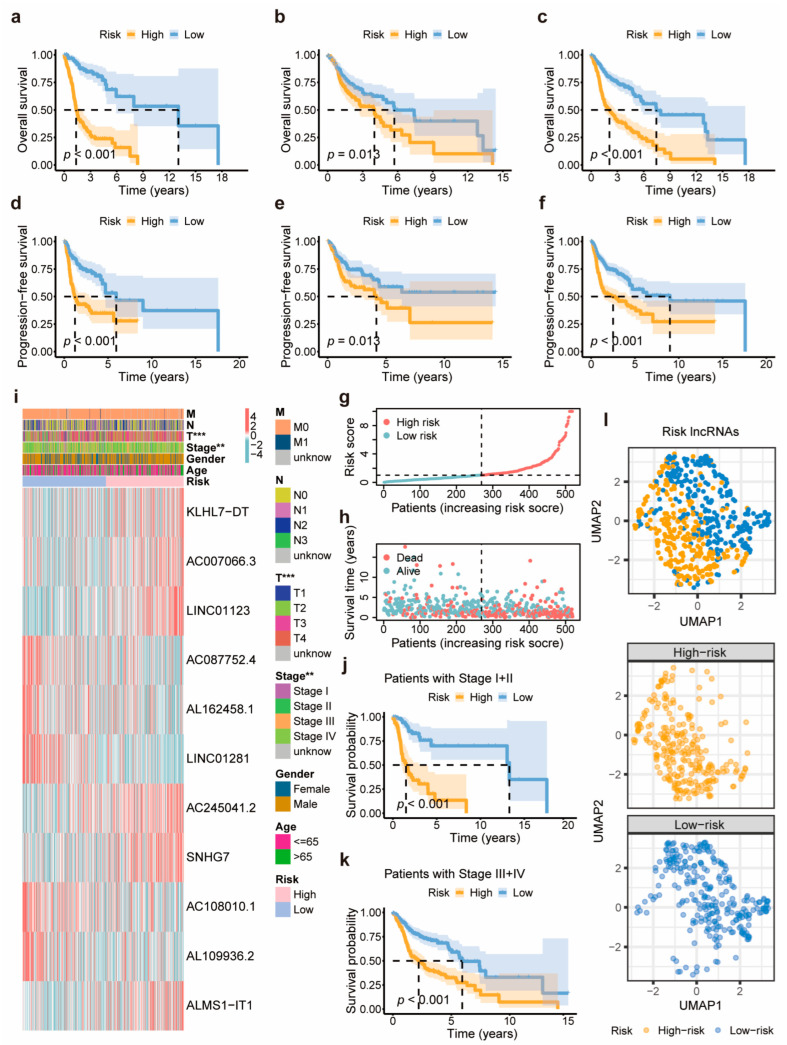
Validation of the 11-DRL prognostic signature. (**a**–**c**) The overall survival of the training group (**a**), the test group (**b**), and the overall group (**c**) through a Kaplan–Meier survival analysis. (**d**–**f**) The PFS of the training group (**d**), the test group (**e**), and the overall group (**f**) through a Kaplan–Meier survival analysis. (**g**–**i**) The risk curve (**g**), survival status (**h**), clinicopathologic heatmap, and 11 model DRL expression heatmap (**i**) of overall TCGA-HNSC patients. (**j**,**k**) Survival curves of TCGA-HNSC patients in the early (**j**) and advanced stages (**k**) between low- and high-risk groups. (**l**) The graphs of UMAP show the clustering situation of low- and high-risk cohorts based on the 11-DRL risk score. **, *p* < 0.01; ***, *p* < 0.001.

**Figure 4 biomolecules-14-00266-f004:**
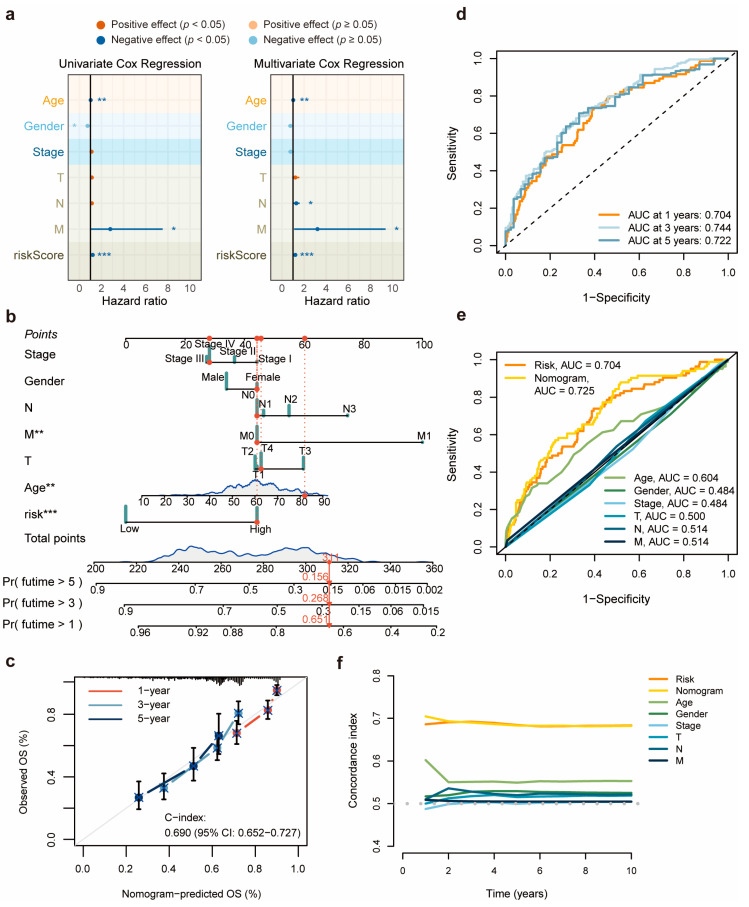
Independent prognostic analysis of risk score and establishment of a nomogram. (**a**) Univariate and multivariate Cox regression analyses of risk score as a prognostic factor in TCGA-HNSC patients. (**b**) The establishment of a nomogram using the clinical characteristics and risk score of TCGA-HNSC patients as elements. (**c**) Calibration curve to evaluate the accuracy of the nomogram in predicting the survival probability of HNSCC patients. (**d**) ROC curves of risk score for the 1-, 3- and 5-year survival probabilities. (**e**) ROC curves of risk score, the nomogram, and other clinicopathological indicators. (**f**) C-index curves of risk score, the nomogram, and other clinicopathological variables. Data are presented as the means with 95% CI. *, *p* < 0.05; **, *p* < 0.01; ***, *p* < 0.001.

**Figure 5 biomolecules-14-00266-f005:**
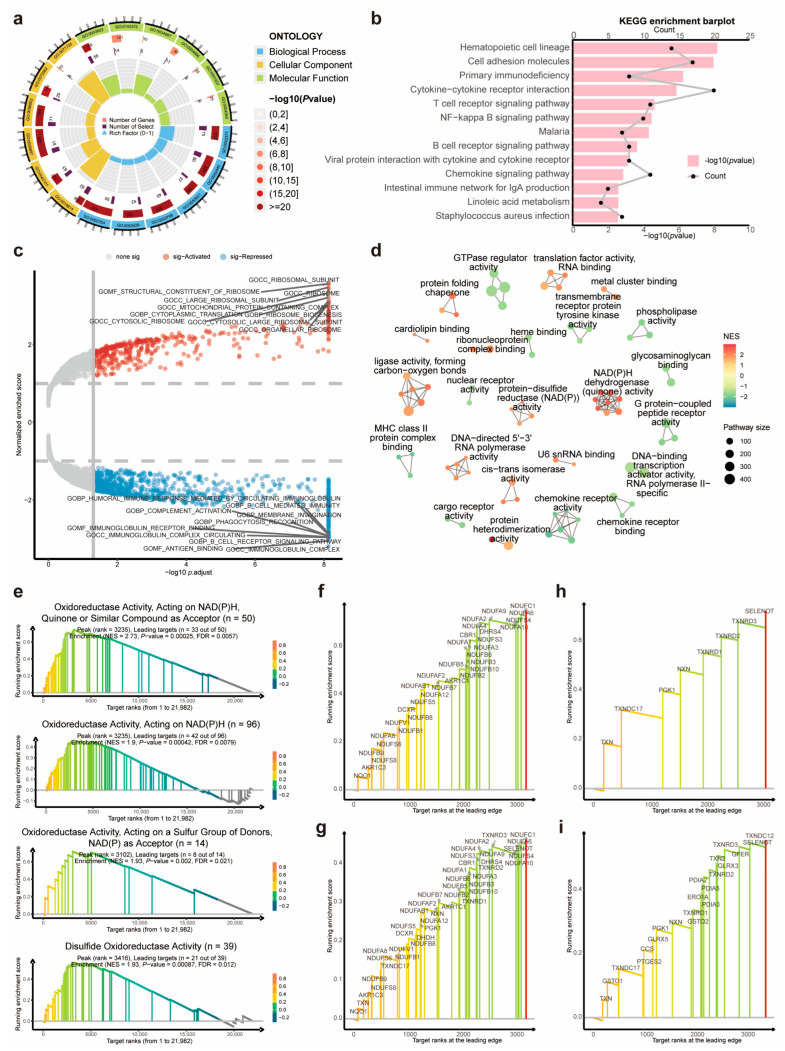
Dysregulated immunity-, NADPH- and disulfide-related pathways between low- and high-risk clusters. (**a**–**c**) GO enrichment analysis (**a**), KEGG pathway analysis (**b**), and the volcano plot of GSEA (**c**) of differentially expressed genes between low- and high-risk clusters. (**d**) The network of similar signaling pathways that were enriched between two risk clusters. (**e**) The enriched signaling pathways that were related to NADPH and disulfide activities. (**f**–**i**) The core genes of the four enriched signaling pathways aforementioned in (**e**): oxidoreductase activity, acting on NAD(P)H, quinone or similar compounds as an acceptor (**f**); oxidoreductase activity, acting on NAD(P)H (**g**); oxidoreductase activity, acting on a sulfur group of donors, NAD(P) as an acceptor (**h**) and disulfide oxidoreductase activity (**i**), respectively.

**Figure 6 biomolecules-14-00266-f006:**
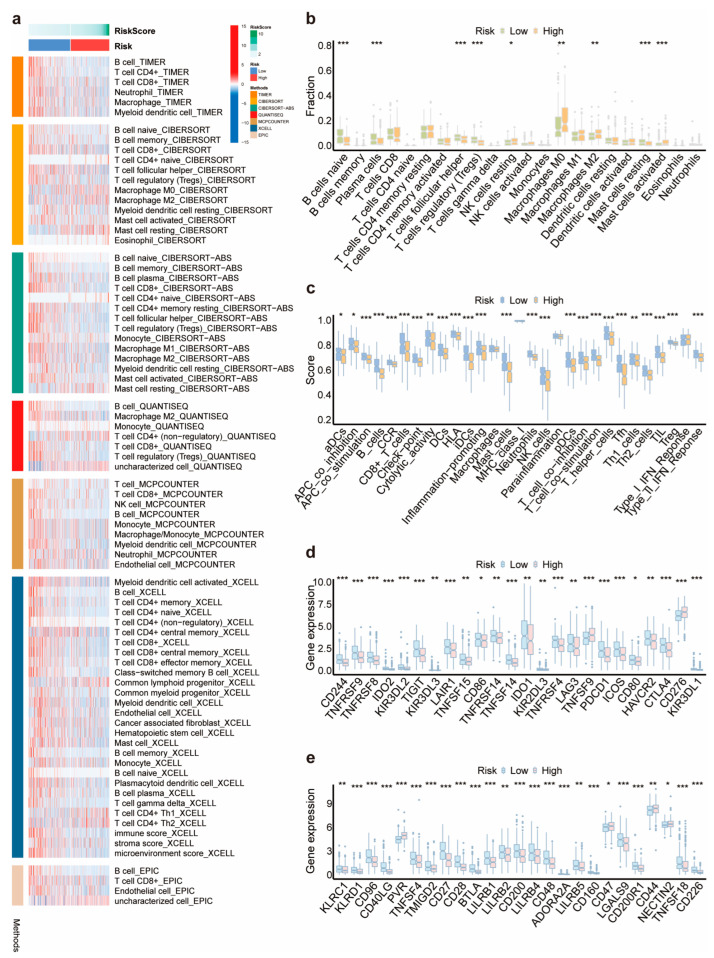
Landscape of immune status in HNSCC and its association with the 11-DRL risk score. (**a**) Correlation graphs of immune cells, stromal cells, and uncharacterized cells in the tumor microenvironment between low- and high-risk cohorts using TIMER, CIBERSORT, CIBERSORT-ABS, quanTIseq, MCP-counter, xCell, and EPIC. (**b**) The difference in immune cell infiltration between the two risk cohorts using CIBERSORT. (**c**) The difference in immune-related functions between the two risk cohorts. (**d**,**e**) The difference in checkpoint expression between the two risk cohorts. *, *p* < 0.05; **, *p* < 0.01; ***, *p* < 0.001.

**Figure 7 biomolecules-14-00266-f007:**
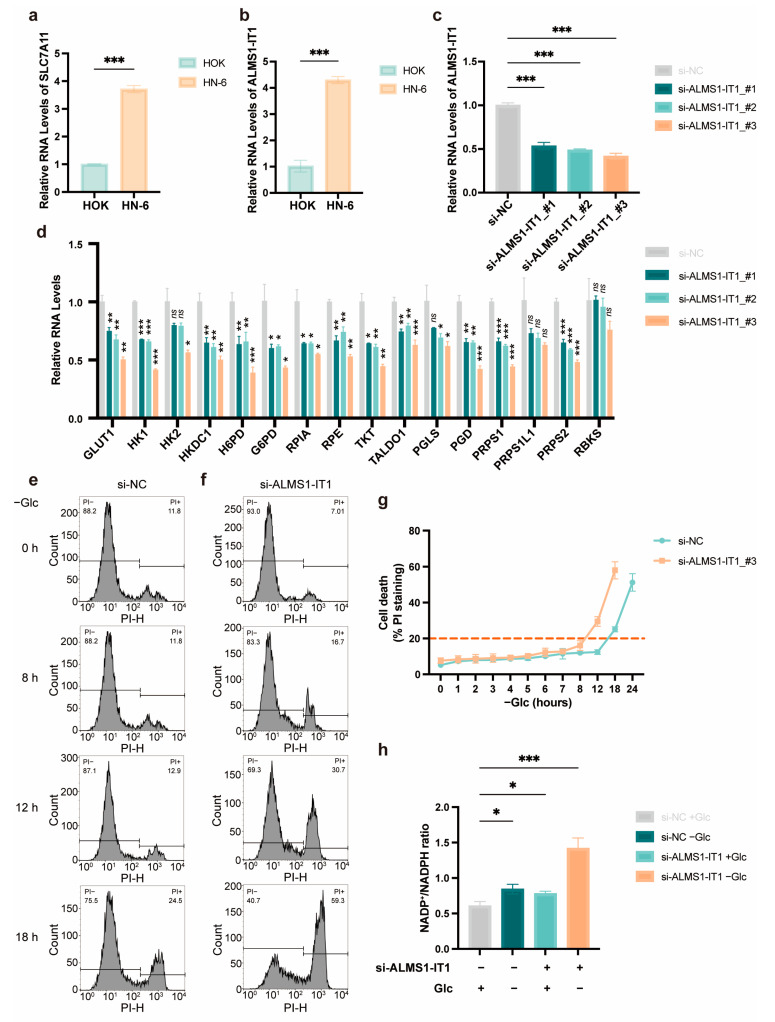
*ALMS1-IT1* modulated cell death under glucose starvation in a PPP-dependent manner. (**a**) The expression levels of *SLC7A11* in HOK and HN-6 cells (normalized with RPL13A). (**b**) The expression levels of *ALMS1-IT1* in HOK and HN-6 cells (normalized with ACTB). (**c**) The knockdown efficiencies of si-ALMS1-IT1_#1, #2, and #3 in HN-6 cells. (**d**) The expression levels of PPP-related genes after *ALMS1-IT1* knockdown. (**e**) Histograms show the ratios of PI+/PI− cells in negative control (si-NC) HN-6 cells at 0 h, 8 h, 12 h, and 18 h after culturing in glucose-free medium, respectively. (**f**) Histograms to show the ratios of PI+/PI− cells in si-ALMS1-IT1_#3 HN-6 cells at 0 h, 8 h, 12 h, and 18 h after culturing in glucose-free medium, respectively. (**g**) The curves of cell death proportions for si-NC and si-ALMS1-IT1_#3 HN-6 cells after culturing in glucose-free medium. (**h**) The NADP^+^/NADPH ratios with treatments of glucose starvation, *ALMS1-IT1* knockdown or both in HN-6 cells. Data are presented as means ± SD. ns, not significant; *, *p* < 0.05; **, *p* < 0.01; ***, *p* < 0.001.

**Figure 8 biomolecules-14-00266-f008:**
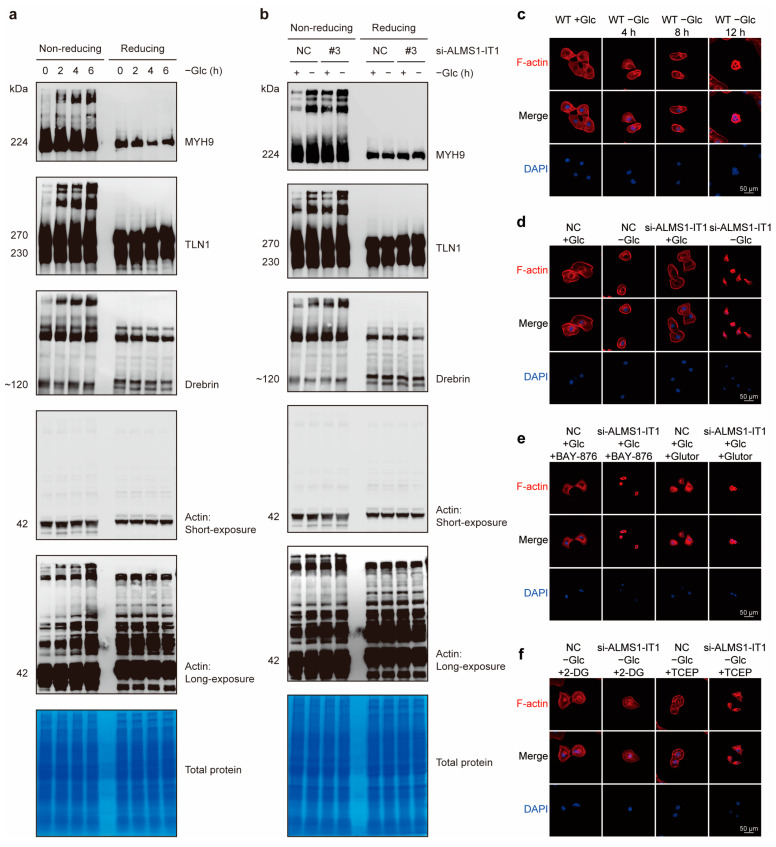
*ALMS1-IT1* regulated disulfide bond formation between actin cytoskeleton proteins and F-actin collapse in disulfidptosis. (**a**,**b**) The Western blot results under non-reducing and reducing conditions of indicated actin cytoskeleton proteins in HN-6 cells with different treatments. (**a**) WT cells were cultured in glucose-containing medium or glucose-free medium for 2 h, 4 h, and 6 h. (**b**) si-NC and si-ALMS1-IT1_#3 cells were cultured in glucose-containing medium or glucose-free medium for 4 h. (**c**–**f**) Fluorescent staining of F-actin with phalloidin in HN-6 cells with different treatments. (**c**) WT cells were cultured in glucose-containing medium or glucose-free medium for 4 h, 8 h, and 12 h. (**d**) si-NC and si-ALMS1-IT1_#3 cells were cultured in glucose-containing medium or glucose-free medium for 8 h. (**e**) si-NC and si-ALMS1-IT1_#3 cells were cultured in glucose-containing medium with BAY-876 (5 μM) or Glutor (5 μM) for 8 h. (**f**) si-NC and si-ALMS1-IT1_#3 cells were cultured in glucose-free medium with 2-DG (2 mM) or TCEP (1 mM) for 8 h. All Western blots and immunofluorescence assays were repeated at least twice independently with similar results. Original figures can be found in Appendix A.

## Data Availability

The datasets analyzed in this study are available at the TCGA repository (https://portal.gdc.cancer.gov/repository (accessed on 29 March 2023)).

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
