# Peer review of "ALMS1-IT1: A Key Player in the Novel Disulfidptosis-Related LncRNA Prognostic Signature for Head and Neck Squamous Cell Carcinoma"

_biomolecules, 2024, doi:10.3390/biom14030266_

Round 1

Reviewer 1 Report

Comments and Suggestions for Authors

In this work, the authors established and validated a novel risk score model composed of 11 disulfidptosis-related lncRNAs (DRLs) based on 24 disulfidptosis-related genes 20 (DRGs) in head and neck squamous cell carcinoma (HNSCC). Furthermore, they demonstrated that knockdown of ALMS1-IT1 inhibited the PPP, contributing to a decline in NADPH levels, which resulted in the formation of multiple intermolecular disulfide bonds between actin cytoskeleton proteins and the collapse of F-actin in the cytoplasm. This study is interesting and novel, which is the first to identify an individual lncRNA that can be regarded as a regulatory factor in disulfidptosis. The manuscript was well-written and clearly presented. 

Below are suggestions:

1#. Since disulfidptosis occurs in SLC7A11 high expression cells, how is the correlation between risk score and SLC7A11 expression in HNSCC patients? Is there positively correlation between them or is SLC7A11 expression high in high-risk group than in low-risk group?

2#.  According to univariate cox regression analysis, there was no statistically significant in N stage, however, it was statistically significant after multivariate cox regression analysis. Could you explain that?

3#. Please label clearly in some figures, for example, in Figure 6e or 6f, figure legend showed that “e Histograms to show the ratios of PI+/PI- cells in negative control (si-NC) HN-6 cells at 0 h, 4 h, 8 h, 12 h and 18 h after cultured in glucose-free medium, respectively.”, but there were only 4 histograms.  

Reviewer 2 Report

Comments and Suggestions for Authors

In the article entitled “ALMS1-IT1: a Key Player in the Novel Disulfidptosis-related LncRNA Prognostic Signature for HNSCC”, the authors investigated disulfidptosis, a newly discovered form of programmed cell death, and the factors involved in the modulation of disulfidptosis-related genes, in head and neck squamous cell carcinoma (HNSCC). They validated 11 disulfidptosis-related lncRNAs (DRLs) based on 24 disulfidptosis-related genes (DRGs) in HNSCC. In particular, they verified the involvement of ALMS1-IT1, one of the 11 model DRLs, in the disulfidptosis of HNSCC cell lines. They verified that ALMS1-IT1 is highly expressed in SLC7A11 high expressed cells, and may be considered a promising therapeutic target for disulfidptosis-focused treatment strategies for cancer and other diseases.

Data reported are interesting and the manuscript is well written, in order to be considered for publication. I think the manuscript may be accepted following some little revisions. I reported here my comments:

The authors should add an ABBREVIATIONS Section to the manuscript in order to clarify the meaning of such acronyms reported within the text.

The authors should add a GRAPHICAL ABSTRACT to the manuscript in order to clarify the hypothesis and the results obtained.

The authors should re-order TABLES sequence number, in particular those of the Supplementary Tables, since they are not ordered following manuscript text.

In the RESULTS Section, sub-paragraph named “3.8. ALMS1-IT1 modulates F-actin collapse during disulfidptosis”, the authors reported that “Silencing ALMS1-IT1 further suppressed the contraction of F-actin. ALMS1-IT1 knockdown suppressed NADPH production, showing a synergistic effect with glucose deprivation (Fig. 7e)”. Have the authors verified the effects of ALMS1-IT1 silencing/knockdown upon the gene expression downstream? It would be interesting to report this data to strongly support authors conclusions.

Please verify paragraphs list numbers. It has been reported “3.2. Figures, Tables and Schemes” but I think it is not the right paragraph number.

In FIGURE 6a-b, the authors have to remove “HN-6” above the graphs.

In FIGURE7c-d-e-f, subpanels have to be enlarged since it is very difficult to see differences within the images.

In conclusion, data are interesting and may be taken in consideration for publication.

The paper needs to be slightly revised in its content in order to meet the journal aims and be considered for publication.

Comments on the Quality of English Language

No comments

Reviewer 3 Report

Comments and Suggestions for Authors

biomolecules-2812953

Title: ALMS1-IT1: a Key Player in the Novel Disulfidptosis-related LncRNA Prognostic Signature for HNSCC

Authors: Sun et al.

Here, the authors addressed a very relevant and innovative topic, namely a novel form of programmed cell death, the disulfidptosis. By bioinformatics approaches the authors analysed relevance of disulfidptosis for head and neck cancer (HNC). Based on disulfidptosis-related lncRNAs they suggest a risk score model to allow prognostic evaluation of HNC. They verified proposed results also in vitro by using a HNC cell line.

In summary, this very comprehensive study is very relevant for the field showing for the first time the relevance of disulfidptosis in HNC. Thus, I suggest acceptance of the article after minor revisions.

Minor comments

-      Why did the authors choose HN6 cells for their analyses? Regarding the high heterogeneity of HNC, it would be important to confirm the results also in at least a second HNC cell line

 -       Abstract: please revise abbreviations; “PPP” should be explained; “DRG” is not used a second time  and can be introduced later in the text
